# Proteomic Analysis in Valvular Cardiomyopathy: Aortic Regurgitation vs. Aortic Stenosis

**DOI:** 10.3390/cells12060878

**Published:** 2023-03-11

**Authors:** Theresa Holst, Johannes Petersen, Sabine Ameling, Lisa Müller, Torsten Christ, Naomi Gedeon, Thomas Eschenhagen, Hermann Reichenspurner, Elke Hammer, Evaldas Girdauskas

**Affiliations:** 1Department of Cardiothoracic Surgery, Augsburg University Hospital, Stenglinstraße 2, 86156 Augsburg, Germany; 2Department of Cardiovascular Surgery, University Heart and Vascular Center Hamburg, Martinistraße 52, 20246 Hamburg, Germany; 3German Center of Cardiovascular Research (DZHK), Partner Site Hamburg/Kiel/Lübeck, 20246 Hamburg, Germany; 4Interfaculty Institute for Genetics and Functional Genomics, University Medicine Greifswald, Felix-Hausdorff-Straße 8, 17487 Greifswald, Germany; 5German Center of Cardiovascular Research (DZHK), Partner Site Greifswald, 17475 Greifswald, Germany; 6Institute of Experimental Pharmacology and Toxicology, University Medical Center Hamburg-Eppendorf, Martinistraße 52, 20246 Hamburg, Germany

**Keywords:** aortic valve disease, valvular cardiomyopathy, chronic heart failure, aortic regurgitation, aortic stenosis

## Abstract

Left ventricular (LV) reverse remodeling after aortic valve (AV) surgery is less predictable in chronic aortic regurgitation (AR) than in aortic stenosis (AS). We aimed to disclose specific LV myocardial protein signatures possibly contributing to differential disease progression. Global protein profiling of LV myocardial samples excised from the subaortic interventricular septum in patients with isolated AR or AS undergoing AV surgery was performed using liquid chromatography–electrospray ionization–tandem mass spectrometry. Based on label-free quantitation protein intensities, a logistic regression model was calculated and adjusted for age, sex and protein concentration. Web-based functional enrichment analyses of phenotype-associated proteins were performed utilizing g:Profiler and STRING. Data are available via ProteomeXchange with identifier PXD039662. Lysates from 38 patients, including 25 AR and 13 AS samples, were analyzed. AR patients presented with significantly larger LV diameters and volumes (end-diastolic diameter: 61 (12) vs. 48 (13) mm, *p* < 0.001; end-diastolic volume: 180.0 (74.6) vs. 92.3 (78.4), *p* = 0.001). A total of 171 proteins were associated with patient phenotype: 117 were positively associated with AR and the enrichment of intracellular compartment proteins (i.e., assigned to carbohydrate and nucleotide metabolism, protein biosynthesis and the proteasome) was detected. Additionally, 54 were positively associated with AS and the enrichment of extracellular compartment proteins (i.e., assigned to the immune and hematopoietic system) was observed. In summary, functional enrichment analysis revealed specific AR- and AS-associated signatures of LV myocardial proteins.

## 1. Introduction

Aortic stenosis (AS) is characterized by progressive narrowing of the aortic valve (AV) orifice area and induces left ventricular (LV) pressure overload [1]. In contrast, aortic regurgitation (AR) is defined by inadequate AV closure during diastole and results in reverse blood flow through the AV, thus leading to LV volume overload [2].

Volume and pressure overload due to chronic AV dysfunction induce LV concentric or eccentric remodeling, eventually resulting in valvular cardiomyopathy and progressive chronic heart failure (CHF), if left untreated [3]. Therefore, AV surgery (repair or replacement) is recommended if symptoms occur or if there is echocardiographic evidence of LV dysfunction [4]. Yet, even after successful AV surgery, a substantial subset of patients experiences persistent or even progressive cardiomyopathy, presumably due to irreversible myocardial dysfunction before surgery [5,6,7]. Clinical observations indicate that this phenomenon is more common in AR patients [6,8,9,10]. However, the precise pathophysiological mechanisms underlying different responses to relief from chronic pressure or volume overload in valvular cardiomyopathy are still insufficiently understood. Currently, no functional or imaging biomarkers are available to predict postoperative lack of LV reverse remodeling in AR patients [10,11].

Profiling of full sets of proteins present in a specific tissue is known as proteomics [12]. Tandem mass spectrometry allows a non-biased analysis of hundreds to thousands of protein levels per sample by measuring peptides derived by proteolytic protein digests. Hence, the relative quantitation of peptides and thereby proteins can help to understand the differences in protein composition, distribution and concentration between diseased and healthy biological systems [13,14]. The approach has been applied to gain better insight into disease-related changes to the protein level in different cardiovascular diseases. For instance, it has been implemented in dilated and hypertrophic cardiomyopathy, atherosclerosis and ischemia/reperfusion injury with the aim to identify differentially abundant proteins compared with controls that hold the potential of serving as diagnostic or prognostic biomarkers or as molecular targets for drug therapy [15]. Furthermore, like pharmacogenomics [16], this high-throughput method has also the potential to be applied for drug repositioning for cardiovascular diseases. 

The LV protein profile in patients with valvular heart disease has rarely been studied. With respect to AV disease, only a single study has analyzed the myocardial protein profile after long-lasting AS [17]. To our knowledge, no study has yet focused on patients with chronic AR. We, therefore, aimed to explore the AR-associated LV myocardial proteome and to compare it to the protein profile of AS patients in order to elucidate molecular differences which might contribute to the differential disease progression in both patient subgroups.

## 2. Materials and Methods

This study was approved by the ethics committee of the Hamburg Medical Association, Germany on 21.02.2016 (PV3759). Written informed consent was obtained from all subjects prior to enrollment. From March 2019 until September 2020, all adult patients referred to our institution for elective surgery for severe AV dysfunction (isolated AR or AS) were considered eligible. Patients were excluded if they had mixed AV disease, concomitant mitral or tricuspid valve disease, relevant coronary artery disease (i.e., coronary artery stenosis > 50%) or declined consent. Furthermore, all patients with active AV endocarditis or acute aortic dissection were excluded. In total, 45 patients (AR: n = 28; AS: n = 17) were included and served as our study population. 

Preoperative work-up included transthoracic and transesophageal echocardiography and routine blood work. AV repair was performed in most patients with AR while AS patients underwent prosthetic valve replacement or the Ross procedure. Independent of the surgical procedure, commissural traction sutures were used for exposure of the AV and underlying left ventricular outflow tract (LVOT). At approximately 1 cm below the AV, a small LVOT myocardial sample was excised from the subaortic interventricular septum in the area underneath the right/left commissure. After excision, the tissue sample was snap-frozen in liquid nitrogen and transported to the laboratory where it was temporarily stored at −80 °C until further processing. 

Proteins were extracted by bead mill (Braun, Melsungen, Germany), operated at 2600 rpm for 2 min and homogenates were reconstituted in 8 M urea/2 M thiourea. Nucleic acid was degraded by benzonase (6.5 U/µg protein, Pierce, ThermoFischer, Langenseibold, Germany). The homogenates were centrifuged at 16,000× *g* for 1 h at 20 °C. Protein concentration was determined by Bradford assay (Biorad, Munich, Germany) with bovine serum albumin as standard. Samples were prepared in multiple batches and measured in parallel with 1 global and 1 batch standard per subset. A total of 5 µg of protein were reduced, alkylated and digested with endoproteinase LysC (Promega, Walldorf, Germany) for 3 h and subsequently with trypsin (Promega) overnight at 37 °C. The reaction was stopped by the addition of acetic acid (final concentration 1%) and peptides desalted on µC18 ZipTip material (Merck, Darmstadt, Germany) according to the manufacturer’s protocol.

Seven outliers were detected during the preliminary quality assessment of lysates from all harvested tissue samples, presumably due to insufficient tissue quality. The final sample set for further, more detailed analysis, hence, comprised lysates from 38 patients in total, including 25 (66%) AR samples and 13 (34%) AS samples (Appendix A). Global protein profiling was performed using liquid chromatography–electrospray ionization–tandem mass spectrometry (LC-ESI-MS/MS, for details, see Appendix A), coupling high-performance liquid chromatography (UltiMate 3000 UHPLC system, Thermo Scientific, Bremen, Germany) and tandem mass spectrometry (Orbitrap Exploris 480 Mass Spectrometer, Thermo Scientific). High-performance liquid chromatography was carried out with a 25 cm Accucore 150-C18 column (inner diameter 75 µg, particle size 2.6 µg, pore size 150 Å, Thermo Scientific) at a flow rate of 300 nL/min. Peptides were separated in a linear 60 min gradient from 7–25% acetonitrile in 0.1% acetic acid, the total run time being 95 min. Data were acquired in data-independent mode. Detailed parameters are compiled in Appendix A. Peptide and protein identification as well as quantitation (i.e., extraction of protein intensities) across the sample set were achieved by using the software package Spectronaut^®^ (Biognosys, Zürich, Switzerland). Peptide sequences were assigned to spectra by the direct data-independent algorithm and the database Uniprot rel.2022_01, limited to human entries. Search criteria included (1) static modification: carbamidomethylation at cysteine, (2) variable modification: oxidation at methionine and protein N-terminal acetylation and (3) fully tryptic peptides. Ion values were parsed when at least 20% of the samples contained high-quality measured values (q < 0.001). Peptides were assigned to protein groups and protein inference was resolved by the automatic workflow implemented in Spectronaut^®^. Only proteins with at least two identified peptides were considered for further analyses. Data has been median normalized on ion level before being compiled to peptides and protein intensities as label-free quantitation (LFQ) values. 

All *p*-values were considered statistically significant if <0.05. Principal component analysis revealed high variance in peptide and protein composition between samples of the same group while standards run for quality control of sample preparation batches and performance of the LC-MS configuration showed low variance. Consequently, the evaluation of the impact of covariates was deemed necessary. Based on LFQ protein intensities, a logistic regression model was calculated in R Statistical Software Version 4.2 (R Studio 2 February 2022) using the *glm* and *confint. default* functions of the *stats* package (v4.2.2, [18]) for evaluation of protein association with AR or AS phenotypes as well as Odds ratios and confidence intervals (95%). The false discovery rate was controlled by the Benjamini-Hochberg method [19]. Consideration of age, sex and protein concentration as covariates led to a stepwise reduction in data variance. Hence, the logistic regression model was adjusted accordingly and used to identify proteins showing associations with patient classification (as defined by Odds ratio (OR, i.e., the chance to be assigned to the AR or AS group when protein intensity is high) and *p* < 0.075). Lastly, web-based functional enrichment analysis of these proteins was carried out using the public tools g:Profiler Version e106_eg53_p16_65fcd97 (https://biit.cs.ut.ee/gprofiler, accessed on 15 August 2022) and STRING (Search Tool for the Retrieval of Interacting Genes/Proteins) Version 11.5 (https://string-db.org/, accessed on 15 August 2022). Both tools functionally characterize input gene/protein lists (i.e., performing over-representation analysis of any known biological function or pathway against a statistical background of the entire human genome) by using cumulative hypergeometric testing along with correction for multiple testing by a tailor-made algorithm [20] or the method of Benjamini and Hochberg [19] for identification of the most significant terms [21,22,23]. STRING additionally allows for the exploration of protein–protein interconnections by network visualization of the input protein list [22,23]. With respect to baseline patient characteristics, categorical variables are expressed as absolute and relative frequencies and normally distributed continuous variables are presented as median (IQR) throughout the manuscript, unless otherwise specified. Comparisons were made using the chi-square test or Fisher’s exact test, as appropriate, or the Mann–Whitney *U* Test. IBM SPSS Version 27.0 software (IBM Corp., Armonk, New York, USA) was used for these analyses.

The mass spectrometry proteomics data have been deposited to the ProteomeXchange Consortium via the PRIDE [1] partner repository with the dataset identifier PXD039662. 

## 3. Results

### 3.1. Study Population

Preoperative patient characteristics are summarized in Table 1. Compared to the AS cohort, there were significantly more male patients in the AR cohort (25/25 vs. 7/13, *p* < 0.001). As a result of volume overload, AR patients presented with expected significantly larger LV diameters and volumes (end-diastolic diameter: 61 (12) vs. 48 (13) mm, *p* < 0.001; end-diastolic volume: 180.0 (74.6) vs. 92.3 (78.4) mL, *p* = 0.001). Furthermore, LV ejection fraction tended to be lower in the AR cohort (56 (13) vs. 60 (8) %, *p* = 0.091). Surprisingly, preoperative creatinine concentration was higher in the AR group (1.06 (0.29) vs. 0.83 (0.16), *p* = 0.006). Age, distribution of AV morphotype, the severity of symptoms, most comorbidities, permanent medication, perioperative risk profiles and interventricular septal wall thickness in diastole were, however, comparable between both groups. Of note, the concentrations of the protein lysates were also similar in both groups. (Table 1).

### 3.2. Overview of Proteins Associated with AR or AS Phenotypes

Of a total 2865 quantified proteins, 171 showed associations with patient classification as revealed by logistic regression analysis. More detailed information on these proteins is outlined in Appendix A. The OR was >1 in 117 proteins, indicating that increasing protein intensities were associated with the AR phenotype while the remaining 54 proteins showed an OR < 1, implying that decreasing protein levels were associated with the AS phenotype. The results of the logistic regression analyses are also displayed in Figure 1. 

### 3.3. Functional Enrichment Analysis of Proteins Positively Associated with AR Phenotype

Of the 117 proteins positively associated with AR, the top 20 proteins are listed in Table 2. 

Functional characterization of the complete list of proteins (n = 117) (Figure 2 (AR) and Appendix A) revealed primarily an enrichment of intracellular compartment proteins including those involved in the carbohydrate metabolism (e.g., AGL, GLB1, GYG1, GYS1, HEXB, ISYNA1, PDK3, PFKFB2, PGM5, PRKAG2, PYGB), nucleotide metabolism (e.g., ATP5S, PRPS1, PRPSAP1, PRPSAP2) and protein biosynthesis (e.g., EEF1E1, EEF1G, EIF4A2, EIF2B1, GARS, GFM1) as well as proteasome components (e.g., PSMA3, PSMB5, PSMB9, PSMD6, PSMD12). 

High levels of these proteins were characteristic of samples of the aortic regurgitation group. For selected proteins Odds ratios are shown in Figure 3. Protein–protein interconnections of all 117 proteins are visualized in Appendix A with the above-mentioned four most enriched functional clusters being color-coded. 

### 3.4. Functional Enrichment Analysis of Proteins Positively Associated with AS Phenotype

Of the 54 proteins positively associated with AS, the top 20 proteins are listed in Table 3. Functional profiling of the complete list of proteins (n = 54) (Figure 2 (AS) and Appendix A) revealed predominantly an enrichment of extracellular compartment proteins including those assigned to the immune system (e.g., EPRS, HLA-C, LGALS1, USP14), hematopoietic system/angiogenesis (e.g., ARPC5, CA2, CDC42, EMILIN1, HBG1, HNRNPA1, HNRNPA3, LTBP1, MPO, VASP) and anemia in specific (e.g., ANK1, EPB42, SLC4A1, SPTA1). High levels of these proteins permit the discrimination of the samples of the aortic stenosis group from those of the aortic regurgitation group. For selected protein odds ratios are shown in Figure 3. Protein–protein interconnections of all 54 proteins are visualized in Appendix A with the above-mentioned three most enriched functional clusters again being color-coded.

## 4. Discussion

Current ESC/EACTS guidelines recommend AV surgery for severe AR and severe high-gradient AS in symptomatic patients regardless of LV function or in asymptomatic patients with echocardiographic evidence of significant LV dysfunction (i.e., LV ejection fraction ≤ 50%, LV end-systolic diameter > 50 mm) [4]. In many patients, however, AV disease is not diagnosed until severe LV dysfunction occurs. At this point, valvular cardiomyopathy may persist and even progress despite successful surgical treatment of AV disease [5,6,7]. Clinical observations indicate that LV reverse remodeling after AV surgery is less predictable in AR patients suffering from volume overload than in AS patients with a predominant pressure overload [6,8,9,10]. This phenomenon has also been demonstrated in a rat model of long-lasting pressure vs. volume overload in which animals with chronic volume overload had less beneficial functional outcomes [24]. The authors of the above-mentioned study hypothesized that increased wall stress and excessive eccentric cardiomyocyte hypertrophy in response to volume overload might be harmful pathogenetic mechanisms in chronic AR. Opposite to that, the ongoing cardiomyocyte renewal in response to pressure overload leading to an increased wall thickness (i.e., concentric hypertrophy) might be protective against irreversible myocardial dysfunction [24]. However, more precise pathophysiological molecular mechanisms underlying both distinct types of valvular cardiomyopathy are still insufficiently characterized. Further elucidation of such pathogenetic pathways is, however, very appealing as it may support the guidance of valvular intervention and individualized decision-making in valvular heart disease.

Recently, Barbarics and colleagues performed global protein profiling of LV and right atrial myocardium chronically exposed to AS [17] and compared their findings to data from healthy controls published by Doll et al. [25]. Among the differentially abundant proteins, they observed a significant enrichment of extracellular compartment proteins including those associated with cardiac hypertrophy, fibrosis and deposition of extracellular matrix as well as blood supply-associated proteins. The authors speculated that these findings may indicate overregulated neo-vascularization pathways induced by compression of arterioles by hypertrophied cardiomyocytes causing reduced blood and nutritional supply [17,26]. Similarly, we found a notable enrichment of hematopoiesis/angiogenesis-related proteins (e.g., ARPC5, CA2, CDC42, EMILIN1, HBG1, HNRNPA1, HNRNPA3, LTBP1, MPO, VASP), specifically of those associated with anemia/hypoxia (e.g., ANK1, EPB42, SLC4A1, SPTA1) in the proteins being positively associated with AS phenotype. In contrast to the above-mentioned study, we, however, compared our findings to AR patients instead of healthy controls. This fact may account for different findings in our study in that we were unable to demonstrate a positive association of hypertrophy and fibrosis-related proteins and extracellular matrix constituents with AS. We hypothesize that cardiomyocyte hypertrophy and fibrosis may be indices of ongoing myocardial remodeling and therefore may similarly occur in long-lasting pressure- and volume-overloaded ventricles [26,27]. Interestingly, we also found a positive association with AS of proteins assigned to the defense/immune system (e.g., EPRS, HLA-C, LGALS1, USP14). Given the fact that an acute immune response is essential for myocardial healing, this finding may indicate ongoing and well-functioning reparative processes in the AS myocardium [28].

The novelty of our study is that we aimed to compare myocardial protein pattern profiles in AR vs. AS patients and, thus, describe for the first time the AR-associated myocardial proteome. We were able to demonstrate a positive association of intracellular proteins involved in energy production and cellular metabolism with AR, including carbohydrate metabolism (e.g., AGL, GLB1, GYG1, GYS1, HEXB, ISYNA1, PDK3, PFKFB2, PGM5, PRKAG2, PYGB), nucleotide metabolism (e.g., ATP5S, PRPS1, PRPSAP1, PRPSAP2) and protein biosynthesis (e.g., EEF1E1, EEF1G, EIF4A2, EIF2B1, GARS, GFM1). Likewise, proteins assigned to the proteasome system (e.g., PSMA3, PSMB5, PSMB9, PSMD6, PSMD12) were markedly enriched. Interestingly, three of the above-mentioned enriched protein clusters have previously been shown to be associated with early stages of CHF: (1) enhanced proteasome activity potentially indicating an ongoing myocardial compensatory response (while end-stage CHF is associated with markedly decreased proteasome function and accumulation of ubiquitinated/oxidized proteins) [29,30], (2) a shift from fatty acid oxidation towards increased glucose metabolism as energy source presumably due to a reduced mitochondrial capacity [31,32], (3) reduced levels of total adenine nucleotides, including adenosine triphosphate, most likely due to substrate wash-out [33,34]. Based on these findings, we hypothesize that the positive association of the described protein clusters with AR but not AS may reflect ongoing efforts of the early-stage failing heart to replenish its energy pools. From a pathophysiological point of view, these enriched protein clusters may indicate a compensatory “intracellular hypermetabolic myocardial state” in response to the increased energy demands of cardiomyocytes subjected to chronic volume overload. If confirmed by subsequent comparison to healthy myocardium, these findings would demonstrate more severe myocardial dysfunction in AR vs. AS patients and thereby support our clinical observation of the more harmful character of AR disease. 

*Limitations:* Tandem mass spectrometry provides a non-biased view of the protein composition of the LV, but molecules being present only in a few numbers in the cells of the LV tissue might be below the detection limit. For now, it remains uncertain to what extent the observed global protein profile in our AR patients differs from the normal condition in patients with preserved AV and LV function due to a lack of healthy controls. For characterization of the healthy human heart proteome, Doll et al. collected tissue from 16 anatomical regions, including both atria and ventricles, from deceased patients during postmortem autopsies [25] while we analyzed LV myocardial samples from living subjects harvested intraoperatively. Direct comparison of enriched proteins in the in vivo vs. post-mortem setting might be potentially misleading due to post-mortem protein degradation. Furthermore, potential differences in sample preparation protocols and uncontrolled baseline characteristics (e.g., age, sex, concomitant conditions, permanent medication) may complicate interpretation. To overcome this limitation, Barbarics and colleagues thus chose to compare the protein ratios of the atrium and the ventricle [17]. However, we obtained myocardial tissue from a single region only (i.e., left-sided subaortic interventricular septum) and could therefore not follow this approach. To allow for the calculation of ratios, myocardial tissue obtained at the same time and in the same fashion from AS patients and healthy controls with similar baseline characteristics (i.e., by performing normalization against AS tissue) would be required. Therefore, to define the clinical relevance of our preliminary findings and to clarify whether the AR-associated protein signature reflects an early stage of CHF with compensatory “intracellular hypermetabolic” activities, comparative proteome analyses and immunohistological studies of the key proteins are crucial. In the long run, this will hopefully aid in further elucidating the clinical observation that AR-induced cardiomyopathy is more likely to be irreversible and unpredictable as compared to rather “benign” AS-induced cardiomyopathy.

*Clinical implications and translational outlook:* Severe symptomatic aortic valve disease ultimately results in valvular cardiomyopathy if left untreated. Yet, left ventricular dysfunction may already be present long before the onset of symptoms. Conventional diagnostics (e.g., echocardiography, late gadolinium enhancement on cardiac magnetic resonance imaging) often fail to detect the first subtle signs of beginning LV remodeling. To stop the downward spiral of progressive myocardial deterioration via timely AV corrective surgery, early detection of LV remodeling is, however, essential. Our study gives new insights into the pathophysiology of LV remodeling in AR patients. We were able to detect an enrichment of protein clusters commonly associated with early stages of CHF (e.g., enhanced proteasome activity, reduced mitochondrial capacity) which may reflect efforts of the LV myocardium to compensate for subtle remodeling processes. Eventually, our findings and results of further subsequent translational research (e.g., correlating findings from proteomics studies and emerging cardiac magnetic resonance imaging techniques for functional and molecular phenotyping, such as feature-tracking strain analysis to visualize and assess subclinical myocardial dysfunction and 31-phosphorus magnetic resonance spectroscopy to visualize and assess early changes in cardiac muscle energy metabolism [35]) could potentially serve as a reference for identification of biomarkers for determination of disease stage.

## 5. Conclusions

In summary, our study demonstrates different LV myocardial protein profiles in AR vs. AS patients. AS was associated with higher levels of extracellular compartment proteins assigned to hematopoiesis/angiogenesis and tissue healing. AR was associated with higher levels of intracellular compartment proteins related to energy production and cellular metabolism which may perhaps indicate a compensatory “intracellular hypermetabolic myocardial state” in response to increased energy demands. Subsequent proteome analyses are required to further elucidate if these differences in protein composition may correlate with specific pathomechanisms leading to different courses of valvular cardiomyopathy in AR vs. AS patients. 

## Figures and Tables

**Figure 1 cells-12-00878-f001:**
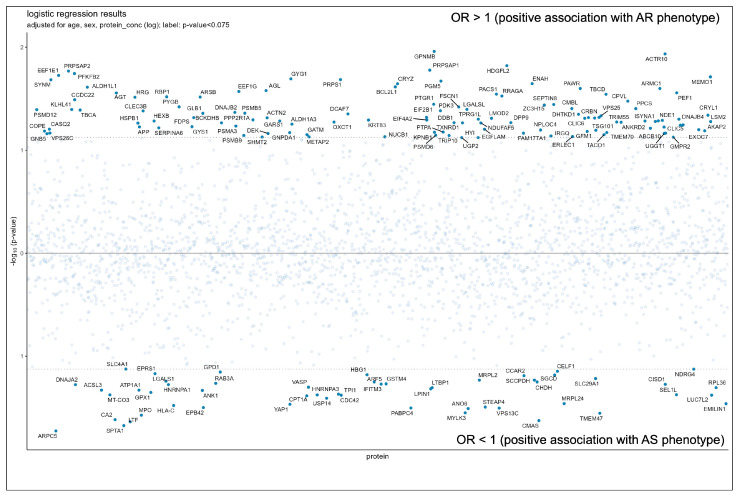
Aortic regurgitation and aortic stenosis-associated proteins: *p* < 0.075. AR: aortic regurgitation; AS: aortic stenosis; OR: Odds ratio, i.e., chance to be assigned to the aortic regurgitation or the aortic stenosis group when protein intensity is high.

**Figure 2 cells-12-00878-f002:**
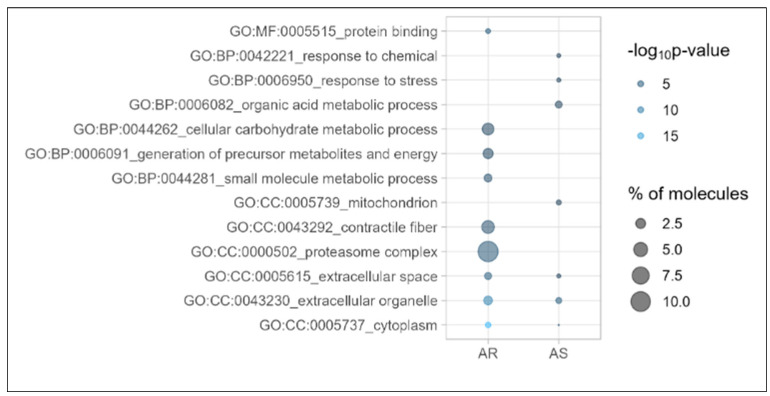
Functional enrichment analysis of list of proteins being positively associated with aortic regurgitation (AR) or aortic stenosis (AS) using g:Profiler. Differential GO terms of levels 2 to 4 shown. BP: biological process; CC: cellular component; GO: gene ontology. −log_10_*p*-value as calculated in g:Profiler with the g:SCS (Set Counts and Sizes) algorithm; % of molecules: percent of molecules per category covered by the protein list analyzed.

**Figure 3 cells-12-00878-f003:**
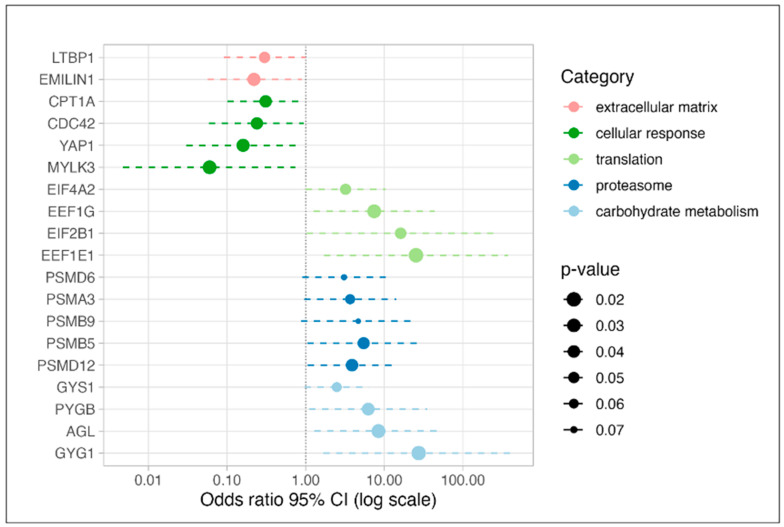
Forest plot of Odds ratios for proteins of specific categories being characteristic of aortic regurgitation (AR; Odds ratio > 1) or aortic stenosis phenotype (AS; Odds ratio < 1). *p*-values and Odds ratio calculated for each protein by the *glm* function of the *stats* package in R using LFQ protein intensities. GO term categorization derived from g:Profiler.

**Table 1 cells-12-00878-t001:** Preoperative patient characteristics.

	Aortic Regurgitation (n = 25)	Aortic Stenosis (n = 13)	*p*-Value *
Age (years)	46 (23)	60 (35)	0.272
Male sex	25 (100%)	7 (54%)	<0.001
NYHA class:			0.912
I	11 (44%)	5 (39%)	
II	8 (32%)	4 (31%)	
III	6 (24%)	4 (31%)	
IV	0 (0%)	0 (0%)	
AV morphology:			0.273
Unicuspid	2 (8%)	4 31%)	
Bicuspid	16 (64%)	6 (47%)	
Tricuspid	7 (28%)	3 (23%)	
Body mass index (kg/m^2^)	25 (4)	26 (7)	0.564
Body surface area (m^2^)	1.98 (0.21)	1.81 (0.36)	0.032
Hypertension	13 (52%)	7 (54%)	0.914
Diabetes	2 (8%)	0 (0%)	0.538
Hyperlipidemia	5 (20%)	1 (8%)	0.643
Coronary artery disease	1 (4%)	2 (15%)	0.265
Permanent medication:			
Beta blocker	12 (48%)	4 (31%)	0.307
ACE inhibitor	10 (40%)	4 (31%)	0.728
AT1-receptor antagonist	1 (4%)	0 (0%)	1.000
Ca^2+^ channel blocker	4 (16%)	0 (0%)	0.278
proBNP (ng/L)	162 (807)	298 (681)	0.484
Creatinine (mg/dL)	1.06 (0.29)	0.83 (0.16)	0.006
EuroSCORE II (%)	0.90 (1.06)	0.90 (1.21)	0.927
Echocardiography:			
LVEF (%)	56 (13)	60 (8)	0.091
LVESD (mm)	40 (8)	30 (1)	0.200
LVEDD (mm)	61 (12)	48 (13)	<0.001
LVESV (mL)	93.4 (63.5)	36.4 (55.5)	0.005
LVEDV (mL)	180.0 (74.6)	92.3 (78.4)	0.001
IVSd (mm)	12 (3)	13 (3)	0.176
Protein concentration (µg/µL)	2.47 (2.97)	2.96 (3.92)	0.649

Data presented as median (IQR) or absolute (relative) frequencies. * *p*-values derived from Mann–Whitney *U* Test (median data) and chi-square/Fisher’s exact test (frequencies). ACE: angiotensin-converting enzyme; AT1: angiotensin 1; AV: aortic valve; Ca^2+^: Calcium; IVSd: interventricular septal wall thickness in diastole; LVEDD: left ventricular end-diastolic diameter; LVEDV: left ventricular end-diastolic volume; LVESD: left ventricular end-systolic diameter; LVESV: left ventricular end-systolic volume; NYHA: New York Heart Association; proBNP: brain natriuretic peptide.

**Table 2 cells-12-00878-t002:** Top 20 proteins with positive association to aortic regurgitation.

Protein ID	Gene Name	Protein Name	*p*-Value *
Q9UBV8	PEF1	Peflin	0.028
Q9NVD7	PARVA	Alpha-parvin	0.052
P46976	GYG1	Glycogenin-1	0.020
O60826	CCDC22	Coiled-coil domain-containing protein 22	0.032
O43324	EEF1E1	Eukaryotic translation elongation factor 1 epsilon-1	0.019
P30153	PPP2R1A	Serine/threonine-protein phosphatase 2A 65 kDa regulatory subunit A alpha isoform	0.051
P25686	DNAJB2	DnaJ homolog subfamily B member 2	0.043
Q14232	EIF2B1	Translation initiation factor eIF-2B subunit alpha	0.048
Q14558	PRPSAP1	Phosphoribosyl pyrophosphate synthase-associated protein 1, PRPP synthase-associated protein 1	0.017
O15061	SYNM	Synemin	0.021
Q15124	PGM5	Phosphoglucomutase-like protein 5	0.021
Q9NZ32	ACTR10	Actin-related protein 10	0.012
Q96IZ0	PAWR	PRKC apoptosis WT1 regulator protein	0.025
O60825	PFKFB2	6-phosphofructo-2-kinase/fructose-2,6-bisphosphatase 2	0.018
P35573	AGL	Glycogen debranching enzyme	0.026
P04196	HRG	Histidine-rich glycoprotein	0.031
Q5T0D9	TPRG1L	Tumor protein p63-regulated gene 1-like protein	0.050
Q16881	TXNRD1	Thioredoxin reductase 1	0.054
O75891	ALDH1L1	Cytosolic 10-formyltetrahydrofolate dehydrogenase	0.024
Q99766	ATP5S/DMAC2L	ATP synthase subunit s, mitochondrial	0.049

* *p*-values calculated by the *glm* function of the *stats* package in R using LFQ protein intensities.

**Table 3 cells-12-00878-t003:** Top 20 of proteins with positive association to aortic stenosis.

Protein ID	Gene Name	Protein Name	*p*-Value *
Q8NFW8	CMAS	N-acylneuraminate cytidylyltransferase	0.024
Q687X5	STEAP4	Metalloreductase STEAP4	0.032
Q709C8	VPS13C	Intermembrane lipid transfer protein VPS13C	0.031
Q32MK0	MYLK3	Myosin light chain kinase 3	0.028
O95573	ACSL3	Fatty acid CoA ligase Acsl3	0.047
O15511	ARPC5	Actin-related protein 2/3 complex subunit 5	0.019
Q9Y383	LUC7L2	Putative RNA-binding protein Luc7-like 2	0.041
P02549	SPTA1	Spectrin alpha chain, erythrocytic 1	0.021
P60174	TPI1	Triosephosphate isomerase	0.043
Q03013	GSTM4	Glutathione S-transferase Mu 4	0.054
P10321	HLA-C	HLA class I histocompatibility antigen, C alpha chain	0.033
Q4KMQ2	ANO6	Anoctamin-6	0.031
P00918	CA2	Carbonic anhydrase 2	0.024
Q92879	CELF1	CUGBP Elav-like family member 1	0.071
Q5T653	MRPL2	39S ribosomal protein L2, mitochondrial	0.059
P07814	EPRS1	Bifunctional glutamate/proline--tRNA ligase	0.068
P46937	YAP1	Transcriptional coactivator YAP1	0.034
Q14693	LPIN1	Phosphatidate phosphatase	0.048
Q8NE62	CHDH	Choline dehydrogenase, mitochondrial	0.056
P51991	HNRNPA3	Heterogeneous nuclear ribonucleoprotein A3	0.042

* *p*-values calculated by the *glm* function of the *stats* package in R using LFW protein intensities.

## Data Availability

Data are available via ProteomeXchange with identifier PXD039662.

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
