# Peer review of "Proteomic Analysis in Valvular Cardiomyopathy: Aortic Regurgitation vs. Aortic Stenosis"

_cells, 2023, doi:10.3390/cells12060878_

Round 1
Reviewer 1 Report
The authors can further discuss the role of cardiac imaging, particularly cardiac magnetic resonance imaging in the non-invasive evaluation of these findings.
In this paper, Holst et al. discuss the proteomic analysis in patients with severe aortic regurgitation and aortic stenosis. They analyzed the proteins in the myocardial sample obtained during the surgery in 25 patients with severe aortic regurgitation and 13 patients with severe aortic stenosis. They have demonstrated different protein profiles among the two diseases, with a higher level of proteins associated with energy production in patients with severe aortic regurgitation.
The findings of this study enhance our understanding of the changes that happen within the myocardium in patients with aortic regurgitation and may help to address the gap in understanding why some patients with aortic regurgitation do not improve after valve replacement. The authors have studied patients with aortic regurgitation, which was not evaluated in prior studies which concentrated on aortic stenosis.
However, discussing the role of advanced cardiac imaging, particularly cardiac MRI, will add to the general application of the finding of this study.
The conclusions were consistent with the evidence and arguments presented and they addressed the main question posed.
The references are appropriate.
Reviewer 2 Report
Within this paper, Holst et al. aimed to detect protein signatures able to discriminate disease progression in aortic regurgitation (AR) vs. aortic stenosis (AS) phenotypes. To do so, they perform protein profiling of AR and AS samples, extract phenotype-associated protein, and perform functional enrichment analysis. The entire procedure revealed specific AR and AS-associated signatures of LV myocardial proteins.
The approach is sound, and the results are novel and convincingly presented (but see my observations on statistics and figures).
I suggest to ACCEPT the paper with MAJOR REVIEWs.
Comment to the authors:
Line 61: Given the fact that the authors propose a brief introduction to proteomics, I would also stress the usefulness of this high-throughput technique in drug repositioning for cardiovascular diseases (see, e.g., PMID:35563496 DOI:10.3390/ijms23095105; PMID:35317720 DOI:10.1186/s12014-022-09345-1; PMID:26768480 DOI:10.1007/s10557-015-6637-y)
Paragraph 134:141 The description of the statistical methods is somewhat "opaque." For example, do the authors use only base R packages (i.e., glm())? If not, they should provide appropriate citations. Finally, are the authors adjusting p-values? If so, how?
I would add a brief phrase in all Tables caption detailing that "p-values are produced by x test (median data) or y test (absolute frequencies).
The caption of Fig.1 is misleading; I would remove the "> 1, .e. chance to be assigned to the [...]."
Lines 207-208 and 225-226 refer to protein positively associated with AR/AS as "favoring classification.", but I do not see any test/evidence of that (e.g., a truth table). Therefore, I would down-tone the affirmation to "were characteristic of." or "permits to discriminate x from y."
Figures
I suggest swapping Figure 2 and Figure 3 labels since the latter is cited earlier within the text (line 208 vs. line 220).
I would like to know how Figure 3 was made: my impression is that the authors used the regression for p-value and OR, and StringDB for the Category code. If that is the case, they should say it in the figure caption, also adding information on how they calculated p-value and CIs.
Figure 2: Would the author explain why they report "-logpBH-value" if in M&M they refer to the g:SCS g:profiler metric (as the "tailor-made algorithm" in ref [17], cited at line 148)?
I am unsure whether the two String networks impact the narrative since they are reported but not explicitly commented on (Figure 4 seems never referred to within the text). I suppose the authors are just using the cluster attribution from StringDB: if so, I would suggest moving the networks to SupMat.
Round 2
Reviewer 2 Report
the paper is acceptable for publication